 eLIFE

# Knockdown of hypothalamic RFRP3 prevents chronic stress-induced infertility and embryo resorption

**Anna C Geraghty[1], Sandra E Muroy[1], Sheng Zhao[2‡], George E Bentley[1,3†], Lance J Kriegsfeld[2,3†], Daniela Kaufer[1,3,4]\*†**

[1]Department of Integrative Biology, University of California, Berkeley, Berkeley, United States; [2]Department of Psychology, University of California, Berkeley, Berkeley, United States; [3]Helen Wills Neuroscience Institute, University of California, Berkeley, Berkeley, United States; [4]Canadian Institute for Advanced Research, Toronto, Canada

**Abstract** Whereas it is well established that chronic stress induces female reproductive dysfunction, whether stress negatively impacts fertility and fecundity when applied *prior* to mating and pregnancy has not been explored. In this study, we show that stress that concludes 4 days prior to mating results in persistent and marked reproductive dysfunction, with fewer successful copulation events, fewer pregnancies in those that successfully mated, and increased embryo resorption. Chronic stress exposure led to elevated expression of the hypothalamic inhibitory peptide, RFamide-related peptide-3 (RFRP3), in regularly cycling females. Remarkably, genetic silencing of RFRP3 during stress using an inducible-targeted shRNA completely alleviates stress-induced infertility in female rats, resulting in mating and pregnancy success rates indistinguishable from non-stress controls. We show that chronic stress has long-term effects on pregnancy success, even post-stressor, that are mediated by RFRP3. This points to RFRP3 as a potential clinically relevant single target for stress-induced infertility.

\*For correspondence: danielak@berkeley.edu

†These authors contributed equally to this work

**Present address:** ‡Department of Biochemistry and Molecular Biology, Southeast University, Jiangsu, China.

**Competing interests:** The authors declare that no competing interests exist.

## Main Text

High psychological stress inhibits reproductive function when both occur concomitantly (*Rivier and Rivest, 1991*; *Ferin, 1999*; *Tilbrook et al., 2000*; *Louis et al., 2011*). From an evolutionary perspective, inhibition of reproductive function by acute stress may be adaptive, delaying reproduction in times of duress or resource scarcity (*Wasser and Barash, 1983*; *Louis et al., 2011*). Chronic stress, however can result in persistent, maladaptive sexual dysfunction and suppressed fertility (*Young et al., 2006*). Little is understood about the lasting effects of stress exposure. For example, after its cessation, can a prior, persistent stressor have long-term negative after-effects on reproductive health? In humans, a high-stress environment may be a significant barrier to sexual well-being and childbearing. In healthy couples under 30 years of age, 63–80% are unable to conceive within 3 months of attempting, and within 1 year of attempting pregnancy, 15% of couples remain unable to conceive (*Practice Committee of the American Society for Reproductive Medicine in collaboration with the Society for Reproductive Endocrinology and Infertility, 2008*). A molecular framework to understand the long-term effects of stress on female reproduction, and its implications for human health, is currently lacking.

The present series of studies sought to answer two main questions: 1) Do stressful events negatively impact female reproductive function even following recovery of the stressor, and, if so, 2) are the deficits observed mediated by stress-induced elevation of the inhibitory neuropeptide, RFamide-related peptide-3 (RFRP3)? To our knowledge, no study to date has elucidated the molecular

**eLife digest** Infertility has become alarmingly common in otherwise healthy women and around 15% of healthy couples younger than 30 years old are unable to conceive within the first year of trying. High-stress levels are known to decrease short-term fertility in humans and other animals, which may serve to prevent pregnancy during times when food or other resources are in short supply.

However, it is not clear if exposure to stress has lasting effects on fertility. Previous studies have found that when male rats experience stress, they release a protein called RFRP3. This protein inhibits brain activity, leading to a reduction in the release of reproductive hormones.

Geraghty et al. took a closer look at how stress may cause lasting fertility problems in female rats. The researchers exposed female rats to stress by restricting their movements for 3 hr each day over the course of 18 days, which increased the levels of stress hormones in the animals. They allowed the rats to recover for one full reproductive cycle—equivalent to a month in humans—and found that while their stress hormone levels returned to normal, RFRP3 levels in the brain remained high. Even after the recovery period, the females were less likely to mate. Also, the females that did mate were less likely to become pregnant, and the ones that did were more likely to lose some of the embryos. Overall, the level of reproductive success in these rats was only 21%, down from 76% in the control group (who were not exposed to the stress).

Next, Geraghty et al. injected a genetically engineered virus into the brain of the stressed rats to switch off the gene that makes RFRP3 during the stress period. This reduced the levels of the RFRP3 protein and restored the mating, pregnancy, and embryo survival rates to the normal levels seen in unstressed rats.

These results suggest that increased levels of RFRP3 during stress can have lasting negative effects on fertility. In the future, developing therapies that lower RFRP3 levels may help individuals who experience fertility problems.

mechanisms of stress-induced infertility nor has there been any investigation of long-lasting after-effects of pre-conception stress on reproductive success and pregnancy outcome. RFRP3, the mammalian ortholog of gonadotropin-inhibitory hormone (GnIH) first identified in Japanese quail (*Tsutsui et al., 2000*), is common across mammals, including rats and mice (*Ukena and Tsutsui, 2001*; *Ukena et al., 2002*; *Kriegsfeld et al., 2006*), hamsters (*Ubuka et al., 2012*), non-human primates (*Ubuka et al., 2009a*), and humans (*Ubuka et al., 2009b*), and is a hypothalamic hormone that directly inhibits the firing of kisspeptin-sensitive gonadotropin-releasing hormone (GnRH) neurons in the hypothalamus in mice (*Ducret et al., 2009*; *Wu et al., 2009*). It also reduces downstream luteinizing hormone (LH) secretion in rats (*Johnson et al., 2007*; *Murakami et al., 2008*), mice (*Son et al., 2012*), and hamsters (*Kriegsfeld et al., 2006*). There is some debate as to whether RFRP3 is a hypophysiotropic hormone (*Murakami et al., 2008*; *Kirby et al., 2009*; *Pineda et al., 2010a, 2010b*) or only centrally inhibits GnRH to elicit a response (*Rizwan et al., 2009, 2012*). Regardless of its mechanism of action, RFRP3 decreases the synthesis and release of pituitary gonadotropins, LH, and follicle stimulating hormone (FSH), in many species, including rats and mice (*Ciccone et al., 2004*; *Johnson et al., 2007*; *Murakami et al., 2008*; *Sari et al., 2009*; *Kriegsfeld et al., 2010*; *Ubuka et al., 2011*; *Son et al., 2012*). In females, RFRP was shown to be regulated throughout the ovulatory cycle in rats and hamsters, and it elicits a marked inhibitory effect on the pre-ovulatory LH surge through inhibition of GnRH activation in rats (*Anderson et al., 2009*). In male rats, RFRP3 expression is elevated 24 hr after a chronic stressor, suggesting that RFRP3 may mediate enduring changes in reproductive function (*Kirby et al., 2009*). Levels of the glucocorticoid stress hormone (in rodents, corticosterone) may mediate this effect; RFRP3 neurons in the rat hypothalamus were shown to express glucocorticoid receptor (GR) (*Kirby et al., 2009*), as well as RFRP-expressing neuronal cell line in vitro (*Lee Son et al., 2014*). Finally, the RFRP promoter region includes two glucocorticoid response elements (GREs), all together supporting the hypothesis that RFRP may be directly regulated by circulating glucocorticoid levels (*Lee Son et al., 2014*). Together, these findings provide support for the notion that stress-induced increases in RFRP3 might have long-lasting negative impact on female reproductive functioning. Despite knowledge of RFRP's responsiveness to stress and its role in regulating reproductive axis

activity, no study to date has established a causal link between RFRP and fertility in any species. We set out to test the potential role of RFRP expression in stress-induced infertility in females.

## Results

In sum, we found that chronic stress led to elevated RFRP3 at all stages of the ovulatory cycle. This elevated level of expression persisted after a full cycle of recovery from stress, indicating that the impact of stress on RFRP3 lasts well beyond removal of the stressor. Stressed females exhibited fewer successful copulation events, fewer pregnancies in those that did successfully mate, and increased frequency of embryo resorption in the achieved pregnancies. These marked effects of stress on fertility were completely blocked by knockdown of RFRP3, even though RFRP3 function was restored following stress cessation. These findings indicate that stress has lingering negative consequences for female reproductive function that are mediated by a transient rise in RFRP3.

Female rats were subjected to an 18 day stress paradigm followed by quantification of hypothalamic markers of reproductive function either immediately after stress exposure or after one full estrous cycle (4 days) of recovery (*Figure 1A*). Serum levels of corticosterone (CORT) were measured on days 1, 4, 7, 11, and 18 of the 18 days immobilization stress paradigm, and on day 22, 4 days after the cessation of stressor. Baseline levels at the onset of stress exposure sessions were unchanged throughout the 18 days. However, CORT levels were significantly elevated in samples drawn on days 1, 4, 7, and 11 at the end of the 3-hr stress exposure.

On day 22, after 4 days of recovery from the stressor, the stressed rats exhibited serum CORT concentrations indistinguishable from baseline values (*Figure 1B*). Rats exhibit a 4–5 days long estrous cycle, with rising estrogen concentrations triggering a surge of luteinizing hormone (LH) to initiate ovulation, and estrogen and progesterone driving sexual receptivity on the night of proestrus (*Blaustein, 2008*). Stress acutely inhibits the LH surge (*Du Ruisseau et al., 1979*) and subsequent sexual receptivity and fertility (*Sirinathsinghji et al., 1983*; *Sato et al., 1996*). However, it is unknown whether reproductive function continues to be negatively impacted even following recovery from stress (defined as exhibiting baseline levels of CORT after 4 days of no stress exposure). Rats were monitored daily by vaginal smear to determine whether estrous cyclicity was affected during application of the stressor and to allow separation of animals into different cycle stages (diestrus, proestrus, and estrus) at the termination of the stressor. Stress did not affect estrous cyclicity, with all animals exhibiting normal vaginal cytology throughout the stressor, and all animals exhibited a normal 4- to 5-day estrous cycle after the cessation of stress. At all estrous cycle stages, RFRP3 mRNA expression in the hypothalamus was significantly elevated both 0 and 4 days after the stressor was terminated (*Figure 1C,D*). Hypothalamic expression of the RFRP3 receptor, G-protein-coupled receptor-147 (GPR147), was also upregulated after stress during all stages of the cycle, and returned to baseline values after the cessation of stress (*Figure 1E,F*). We did not find significant differences in either GnRH or kisspeptin (KISS1) mRNA expression post-stress in any stage of the cycle (*Figure 1G–J*). However, hypothalamic samples were taken from whole hypothalamus, precluding the ability to differentiate between rostral and caudal kisspeptin cell populations, potentially masking subtle differences. Notably, the persistent increases in the expression of both RFRP3 and its receptor specifically in proestrus coincide with the cyclical onset of sexual receptivity, suggesting that RFRP3 provides a mechanistic basis for long-lasting suppression of reproductive behavior after stress.

To investigate whether the stress-induced increase in RFRP3 plays a causal role in prolonged sexual inhibition, we developed a conditional viral vector to knock down RFRP3 expression (tet-OFF lentivirus RFRP3 shRNA) in vivo during the strictly-defined time window of the chronic stressor. This lentiviral construct expresses RFRP3 shRNA from a constitutively active CMV promoter, driving both shRNA and blue fluorescent protein (BFP) marker expression. When exposed to doxycycline (DOX, via drinking water) (*Figure 2A*), the tet-Off element is prevented from driving TRE-initiated transcription and both shRNA and BFP production cease (location and extent of viral infection can be seen in *Figure 2B,C*). Stereotaxic infusion of RFRP3 shRNA lentivirus into the hypothalamus led to an 87% down-regulation of RFRP3 mRNA expression within 7 days relative to a control-scrambled shRNA (*Figure 2D*). Immunohistological labeling verified that the peptide level in the hypothalamus was similarly knocked down by 85% compared to scrambled control virus, measured 2 weeks following viral injection (*Figure 2E*, representative images of RFRP labeling with either scramble or RFRP-shRNA virus and pre- and post-DOX administration *Figure 2F–I*). Critically, administration of doxycycline in the drinking water restored RFRP3 mRNA to normal levels within 4 days (*Figure 2D*). This viral vector

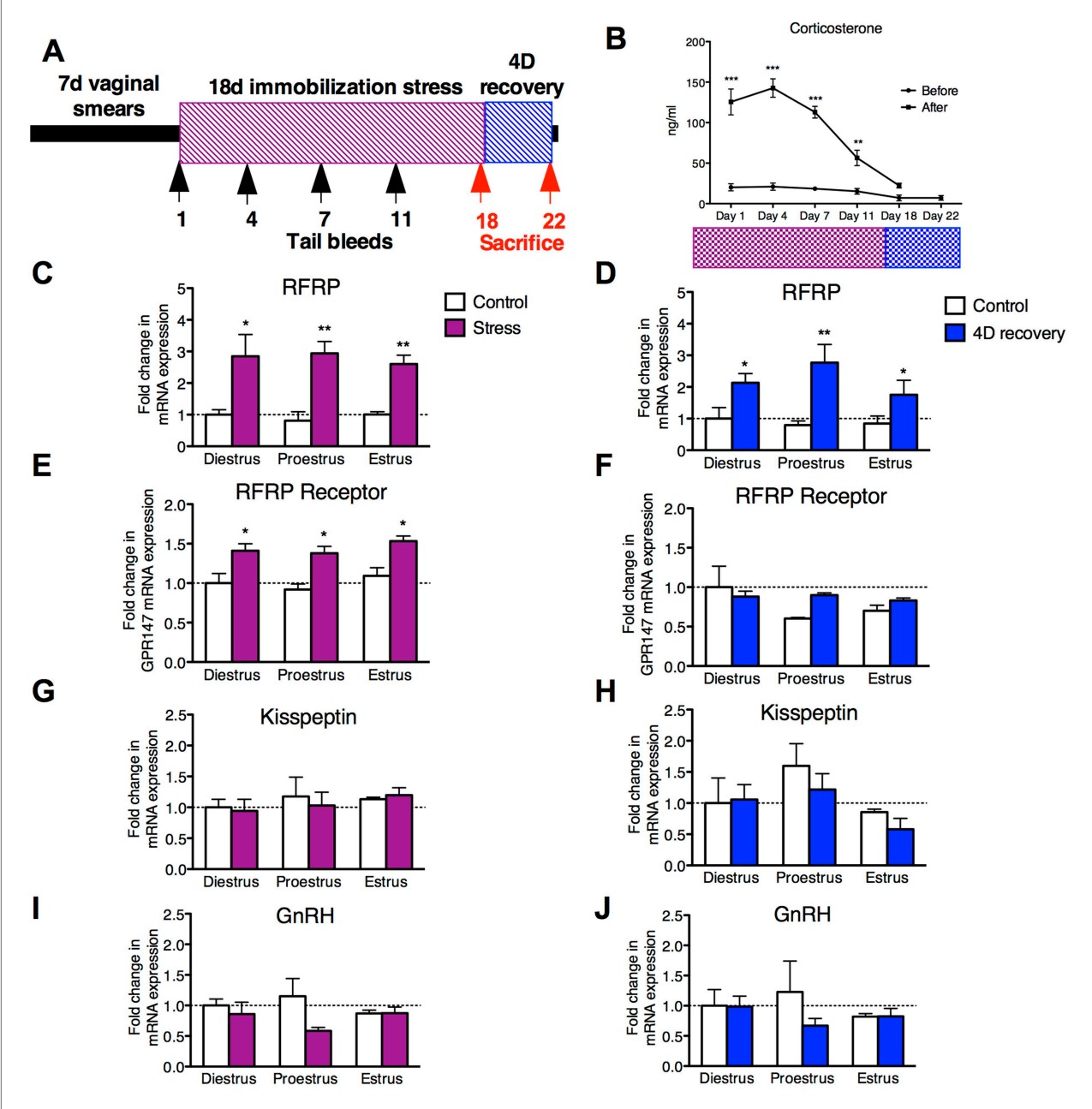

**Figure 1**. 18 days chronic stress leads to an upregulation of RFRP mRNA that persists for at least one estrous cycle in the rat. (**A**) Experimental timeline. (**B**). Corticosterone was measured in serum samples from tail vein blood immediately before and after stress sessions on days 1, 4, 7, 11, and 18, and on day 22, 4 days post-stress cessation (N = 36/group in 1,4,7,11,18 timepoints, N = 18/group on 22). (**C**, **E**, **G**, **I**) Gene expression changes in the hypothalamus immediately after stress and (**D**, **F**, **H**, **J**) 4 days after stress. mRNA levels of all (mean ± SEM, N = 6/group) were determined using qRT–PCR relative to the ribosomal reference gene RPLP at day 0 and 4 post-stress cessation. Estrous cycle staging was determined by inspection of daily vaginal smears. *p < 0.05, **p < 0.01, ***p < 0.001. PCR statistics were done by a Kruskal–Wallis one-way ANOVA followed by Dunn's multiple comparison test for post-hoc analysis, CORT statistics analyzed by a repeated two-way ANOVA.

system permitted knocking down of RFRP3 expression during chronic stress and restoration of RFRP3 during the later stages of copulation, mating, and birthing, which may rely on RFRP3 function in unknown ways.

A second group of female rats received dorsomedial hypothalamic injections of either RFRP3 shRNA or a scrambled control shRNA lentivirus 3 weeks before the 18 days of immobilization. Estrous

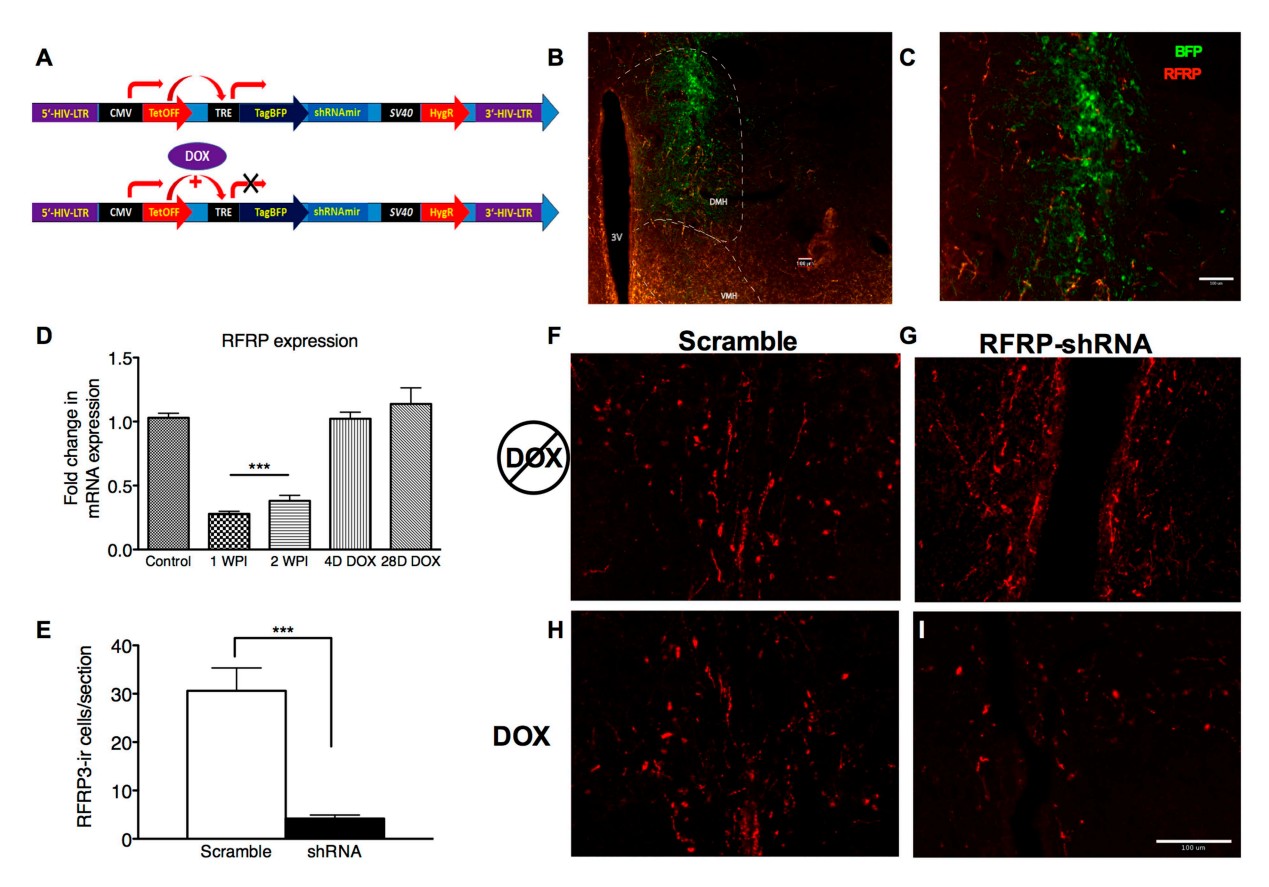

**Figure 2**. RFRP-shRNA successfully knocks down RFRP expression in the dorsal medial hypothalamus, and expression is recovered upon DOX induction. (**A**) Map of RFRP-shRNA viral plasmid. (**B**) Brain sectioned and stained with an anti-BFP antibody to label virus infection (green) and anti-RFRP 2 weeks post-injection (WPI) to show injection location and (**C**) spread. Scale bar indicates 100 μm. (**D**) mRNA levels of RFRP following injection of RFRP-shRNA viral vector were determined using qRT-PCR (WPI = weeks post-injection, mean ± SEM, N = 4). (**E**) RFRP3-ir cells/section counts in the DMH after 2 weeks post-injection with either scramble or RFRP-shRNA virus. (**F–I**) Brain sectioned stained with anti-RFRP3 antibody 2 weeks post-injection with scramble or RFRP-shRNA virus and before and after DOX administration. Scale bar indicates 100 μm. *p < 0.05, **p < 0.01, ***p < 0.001. Statistics were done by one-way analysis of variance (ANOVA) and Bonferroni post-hoc tests. For mRNA data, PCR statistics were done by a Kruskal–Wallis one-way ANOVA followed by Dunn's multiple comparison test for post-hoc analysis and statistics for protein counts were a student's t-test.

cycles were monitored for each rat with immobilization timed to coincide with the onset of estrus, leaving most rats in proestrus 4 days after the end of stress. All rats were administered DOX on the final day of stress so that restoration of RFRP3 expression coincided with the onset of proestrus after the 4-day recovery period. (*Figure 3A*). After one full estrous cycle of recovery from stress, rats underwent a timed mating test on the night of proestrus and were monitored through gestation and birth, to assess the long-term effects of stress on reproductive success including successful copulation and pregnancy outcome.

Tail vein serum samples taken at the onset and end of stress sessions on days 1, 11, and 18 revealed that post-stress circulating levels of CORT were elevated on days 1 and 11 of the immobilization period (*Figure 3B*). Moreover, RFRP3 knockdown during stress did not significantly alter CORT response during stress, indicating an intact hormonal stress response (*Figure 3B*). Stress exposure led to a profound decrease in total reproductive success in females that received the control virus: only 21% of stressed females became pregnant and carried to live birth, as compared to 76% of non-stressed females with control virus (*Figure 3C*). 80% of the females that received the RFRP-shRNA virus became pregnant and carried to live birth (*Figure 3C*). The stress-induced decline in reproductive success resulted from a cumulative decrease in mating success (from 88 and 90% in non-stressed groups to 43% in the stress-scrambled group, *Figure 3D*) and pregnancy rates in the females that

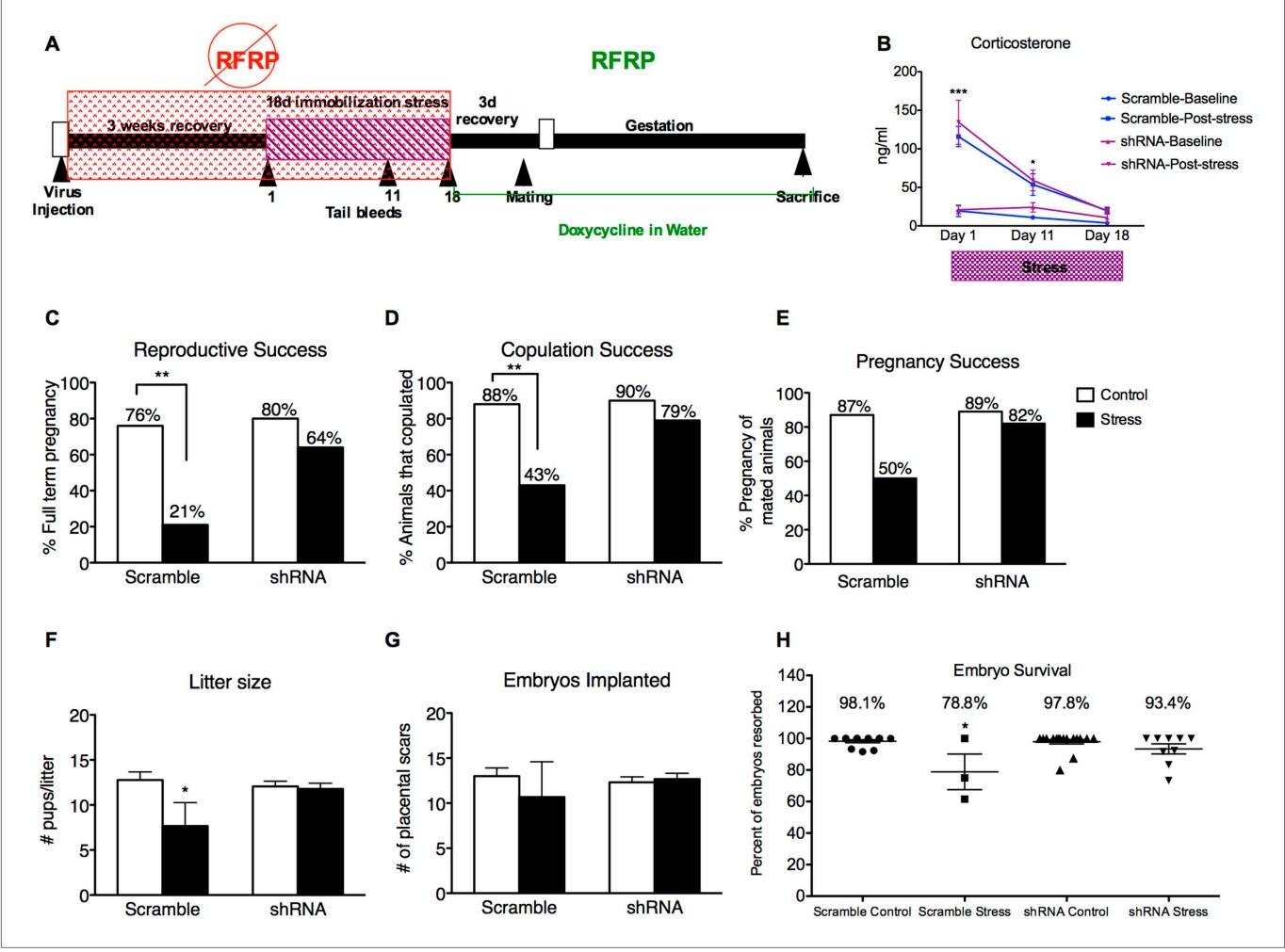

**Figure 3**. Knocking down RFRP during stress completely prevents stress-induced reproductive dysfunction. (**A**) Experimental time line. (**B**) Corticosterone concentrations were measured in serum samples from tail vein blood immediately before and after stress sessions on days 1, 11, and 18. (**C**) Total reproductive success was measured as percentage of females that successfully brought a litter to full term (Scramble/control N = 17, Scramble/Stress N = 14, shRNA/control N = 20, shRNA/stress N = 14, g-statistics: G = 5.836, df = 1 p = 0.016, fisher's exact test p = 0.0031). Breaking down total reproductive success, (**D**) copulation success was measured as percentage of females that exhibited lordosis and allowed a male to achieve intromission within 15 min (g-statistics: G = 2.405, df = 1 p = 0.028, fisher's exact test p = 0.0062), and (**E**) pregnancy success refers to the percentage of females that got pregnant out of the subgroup that successfully copulated (Scramble/control N = 15, Scramble/Stress N = 6, RFRP-shRNA/control N = 18, RFRP-shRNA/stress N = 11). (**F**) Litter sizes measured as number of pups born alive immediately after birth (dams-Scramble/control N = 13, Scramble/Stress N = 3, RFRP-shRNA/control N = 16, RFRP-shRNA/stress N = 9). (**G**) Embryos implanted measured as number of placental scars identified in the dam's uterine horns after birth. (**H**) Embryo survival was calculated as the number of birthed pups divided by number of maternal placental scars and shown as a percentage (indicative of initial implantation, mean ± SEM) *p < 0.05, **p < 0.01, ***p < 0.001. Reproductive success statistics were done by G-statistics tests followed by Fisher's Exact test, statistics for litter size, placental scars, and embryo resorption were done by a two-way analysis of variance (ANOVA) followed by Bonferroni post-hoc tests and CORT statistics analyzed by a repeated two-way ANOVA.

The following figure supplement is available for figure 3:

**Figure supplement 1**. RFRP-shRNA animals had increased plasma estradiol on proestrus during stress and RFRP-shRNA animals that mated had higher circulating estradiol than scrambled animals.

mated (from 87 and 89% in non-stressed groups to 50% in the stress-scrambled group, *Figure 3E*). Interestingly, knockdown of RFRP3 expression in the hypothalamus during stress exposure prevented the stress-induced suppression of reproduction, leading to 79% copulation success, 82% pregnancy success, and overall reproductive success to 64%, a rate statistically equivalent to control (non-stress) levels (76%, *Figure 3C*).

Exposure to acute stress on the evening of the third day of pregnancy was reported to lead to reduced litter size, via inhibition of implantation which occurs normally 5 days after mating (*Zhao et al., 2013*). Therefore, we next assessed whether pre-copulation stress exposure affects pregnancy outcome. Stressed females that received control-scrambled shRNA had significantly smaller litter sizes (*Figure 3F*, 12.77 ± 0.91 vs 7.667 ± 2.60 pups/litter, p < 0.05) with no difference in placental scars (*Figure 3G*, 13.0 ± 0.91 vs10.667 ± 3.93, p > 0.05). Embryo survival was analyzed in the females that were successfully impregnated by determining the ratio of placental scars (indicative of successful implantation) to the number of live pups in the litter. These were first pregnancies for all females, so the number of placental scars is indicative of implantation events during this pregnancy. Embryo survival in stressed females that received scrambled shRNA was 78.8 ± 11.7% of fetuses, compared to 98.1 ± 0.95%, and 97.8 ± 1.5% survival in the control scrambled and control RFRP-shRNA groups, respectively (*Figure 3F*, p < 0.05). Most remarkably, RFRP3 shRNA administration suppressed stress-induced fetal resorption, showing a 93.4 ± 3.2% fetal survival rate (*Figure 3H*). These results demonstrate that stress-induced increases in RFRP3 expression has long-term detrimental effects on female reproductive fitness that persist long after the stressor has been removed and CORT levels have returned to baseline. In addition, knocking-down RFRP3 expression during stress eliminated the stress-induced decrease in sexual motivation, decrease in pregnancy success, and subsequent increase in embryo resorption.

We next examined plasma estradiol concentration in animals throughout the stress period in both scrambled and RFRP-shRNA groups. We found that animals with RFRP knocked down had significantly higher circulating estradiol in proestrus during the stress exposure than animals that received the scrambled virus (*Figure 3—figure supplement 1A*, p < 0.01), indicating that RFRP knockdown reverses the stress-induced blockade of the E2 rise that occurs during proestrus. Examining animals more closely during proestrus, we found that the RFRP-shRNA animals that successfully mated after the stressor had significantly higher circulating estradiol in their proestrus periods over the course of the stressor than both scrambled groups (*Figure 3—figure supplement 1B*, p < 0.05).

Finally, we investigated behavioral measures of female receptivity to test its potential contribution to the stress-induced reproductive deficits observed. Lordosis intensity is a rating (from 0-3) of the quality of all lordosis poses the female exhibits during the mating session, when 0 marks no lordosis and 3 is a fully mounted spinal flexion pose. In rats, a common index of relative sexual receptivity of a female in the presence of males, is the lordosis quotient (LQ), calculated by the number of times the female adopts a lordosis posture scored 2 or higher, divided by the number of times a male mounts her. All females included in the study exhibited lordosis when introduced to a male (indicating that they were in the correct stage of their cycle). Lordosis intensity did not differ within groups (*Figure 3—figure supplement 1C*) but a significant main effect of stress revealed that lordosis intensity was significantly suppressed by stress (F(1, 61) = 5.15, p = 0.0268). Furthermore, lordosis quotient measures revealed significantly lower ratio in the scrambled stress group compared to the non-stressed groups that received scrambled or RFRP-shRNA (0.30 ± 0.10 vs 0.73 ± 0.07 and 0.68 ± 0.07), indicating that stress exposure decreased the relative sexual receptivity of the females (*Figure 3—figure supplement 1D*), congruent with the stress-induced drop in mating success we found. Interestingly, LQ ratios in stressed females that received RFRP-shRNA were not significantly different from controls ratios (0.53 ± 0.10, *Figure 3—figure supplement 1D*), demonstrating that knock-down of RFRP reversed the stress-induced decrease in sexual receptivity, and congruent with the reversal of mating success found in this group.

## Discussion

In humans, high anxiety and psychological stress can lead to long-term impaired fertility, ranging from reduced libido, delayed pregnancy success to the extreme of complete reproductive axis suppression as in the case of hypothalamic amenorrhea (*Ferin, 1999*). In our studies, selective knock-down of hypothalamic RFRP3 during stress exposure preserved all aspects of reproductive function that were otherwise reduced in stress-exposed animals (summary schematic, *Figure 4*). The stress-induced spike in RFRP3 initiates a long-lasting suppression of reproduction, well after the removal of the stressor, perhaps via positive feedback that maintains elevated RFRP3 levels or engages downstream suppressive targets. These findings reveal a single molecular target that persistently underlies a range of different reproductive dysfunctions that may provide a novel translational framework for clinical study of human reproductive health.

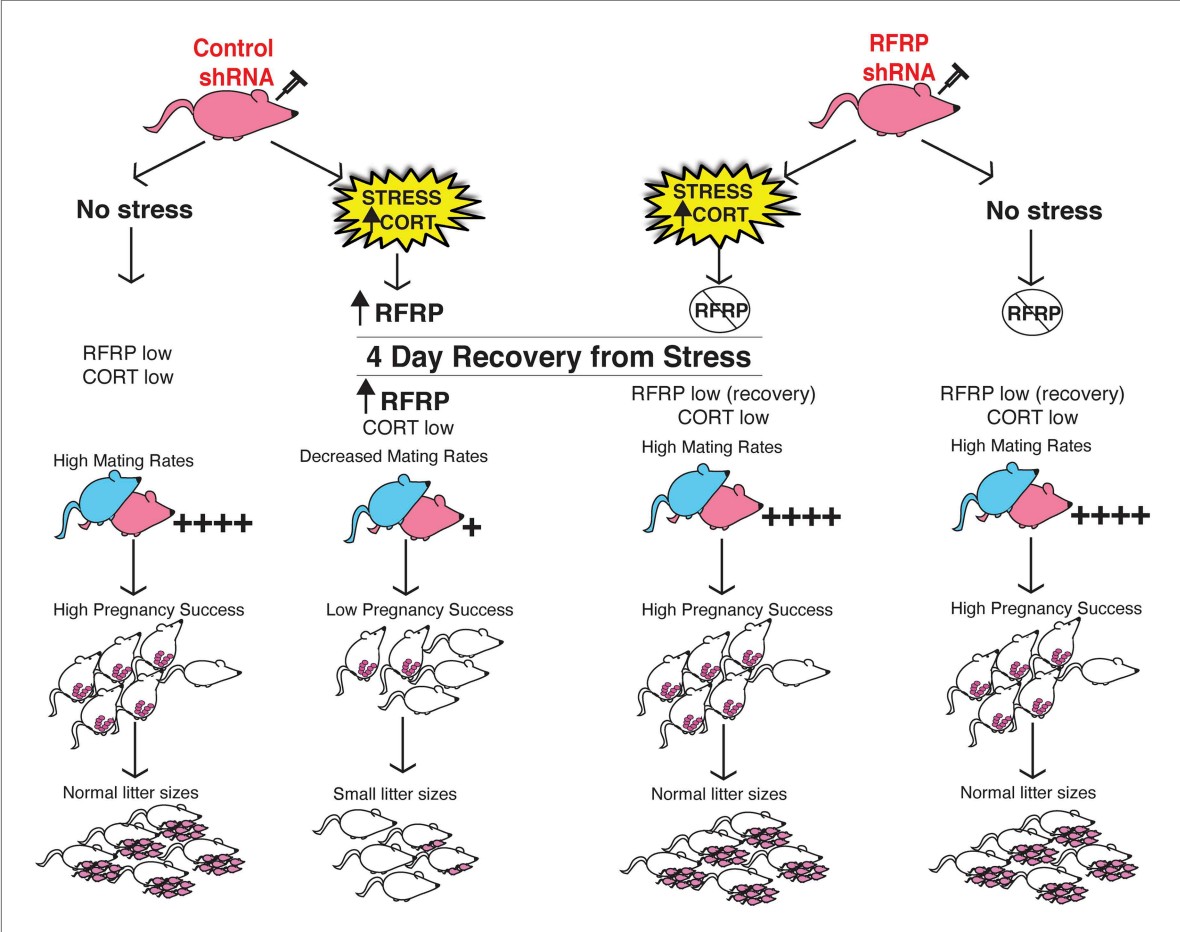

**Figure 4**. Schematic illustration of experiments. Female rats (in pink at top) were injected with either an inducible RFRP-shRNA or a scrambled control virus. Each group was furthered separated into a no stress control or subjected to 18 days of immobilization stress. Stressed females exhibited fewer successful copulation events, fewer pregnancies in those that did successfully mate, and increased frequency of embryo resorption. These marked effects of stress on fertility were completely blocked by knockdown of RFRP3.

The stress-induced rise in RFRP may be acting on neural circuits influencing mating and pregnancy, potentially independently of sex steroids. RFRP projects to multiple brain regions responsible for successful reproduction and mating behavior, including the medial pre-optic area (mPOA) (where it is known to affect GnRH release) as well as the BNST, medial amygdala, anterior hypothalamus, and arcuate nucleus (*Kriegsfeld et al., 2006*). Piekarski et al. found that administering RFRP3 to hamsters reduced sexual motivation (as measured by percent of time spent with castrated vs intact males) and vaginal scent marking without effect on lordosis behavior, similar to our present findings. Additionally, RFRP3 administration altered cellular activation in regions of the brain implicated in female sexual motivation, including the mPOA, medial amygdala, and BNST—all regions that receive RFRP projections. These effects were independent of gonadal steroids and kisspeptin cellular activation (*Piekarski et al., 2013*). While we were unable to measure progesterone or prolactin in this study, it is possible that RFRP projections to the arcuate nucleus affect dopaminergic signaling required for prolactin release and maintenance of progesterone levels and pregnancy success. Future studies aimed at systematically examining each step in these processes is required to gain a full understanding of the neural circuits underlying the deleterious effects of stress on reproduction.

In humans, RFRP 1 and 3, and their cognate receptor are expressed in the hypothalamus (*Ubuka et al., 2009b*). It is possible that manipulation of RFRP3 signaling in humans may relieve stress-related reproductive dysfunction, including decreased sex drive, impaired fertility, and increased miscarriages. Likewise, if similar mechanisms of stress-induced reproductive suppression are common across

species, such strategies may be similarly relevant to species bred in captivity that are susceptible to stress-induced infertility, in particular, endangered species whose preservation depends on captive breeding programs.

## Materials and methods

### Experimental subjects

Adult female Sprague–Dawley rats were triple-housed on a 12/12-hr light–dark cycle with lights on at 0700 hr and ad libitum food and water. For all studies, rats were acclimated for a week and then vaginal smears were obtained daily to verify normal cyclicity for 12 days before the studies commenced. Rats that did not cycle normally were removed from the study. For the chronic stress experiment, rats were immobilized daily from 9 am to 12 pm (N = 6 for each cycle time point) or left undisturbed in their home cages (N = 6 for each cycle time point) until terminal samples were collected (stress paradigm described below). In the RFRP knockdown study, animals received stereotaxic injections of either RFRP-shRNA (N = 30) or scrambled control (N = 28), then allowed to recover for 3 weeks. After recovery, rats were exposed to the same stress paradigm as the previous experiment. After cessation of stress all animals were left undisturbed in their home cage for 4 days, and on the night of the fourth day observed during timed mating (see below). Rats that successfully mated were left in their home cages for the duration of gestation and then perfused within 24 hr after parturition. (stress/shRNA, N = 14, control/shRNA, N = 20, stress/scrambled, N = 14, control/scrambled, N = 17). One cage of control/scrambled animals were removed from analysis due to fighting. All animal care and procedures were approved by the University of California–Berkeley Animal Care and Use Committee (Protocol R303-0313BC).

### Immobilization stress

Rats were immobilized in Decapicone bags (Braintree scientific) and placed in individual cages in a fume hood for 3 hr/day for 18 days from 9am-12pm. Blood samples were collected for corticosterone measurement on days 1, 4, 7, 11, and 18 at the onset of the stressor and again at the end of the 3 hr.

### Plasma corticosterone and E2 hormone sampling

All blood samples were collected from tail vein and centrifuged at 2000×$g$ for 15 min. Plasma was extracted and stored at −20°C until assayed. Corticosterone was measured using a Corticosterone EIA kit (Enzo Life Sciences, Farmingdale, NY) with individual samples used for analysis. Sample values below the detection level of the assay were included as the lowest detectible value. Samples were assayed in duplicate and groups were balanced across different plates. Inter-assay coefficients were <3% and intra-assay coefficients were <5%.

Estradiol assays were run at the University of Virginia Center for Research in Reproduction and were measured by CalBiotech EIA (Spring Valley, CA) in singlet with individual samples used for analysis. Again, sample values below the detection level of the assay were included as the lowest detectible value. Inter-assay coefficients were <2% and intra-assay coefficients were <8%.

### Copulation tests

Females verified to be in proestrus were paired with a novel male in a large rectangular cage under red light illumination during the lights off phase. The male was permitted to mate with the female for up to two ejaculations after which the male was removed from the cage. The interactions were video-taped and both male and female behaviors were blindly scored post-hoc. Females that never exhibited lordosis posture during the test were removed from analysis. Females that exhibit at least one lordosis posture but that did not allow the male to achieve intromission were termed an 'unsuccessful maters' and after 15 min removed from the cage.

### Scoring of sexual behaviors

All mating tests were videotaped in real-time for subsequent behavioral scoring. Videos were scored by two individuals blind to the experimental conditions. Behaviors of male and female animals were scored. The lordosis intensity was scored on a 4-point scale (0–3) as described by Hardy and DeBold (*Hardy and Debold, 1971*) where 0 indicates no lordosis response and 3 indicates a pronounced spinal flexion, and averaged over the number of lordotic poses presented. The lordosis quotient (LQ) was determined as the number of lordosis responses (scores of 2 or 3) divided by the total number of mounts during the scored session. The number of proceptive behaviors was calculated as number of

ear wiggles/minute during duration of test, as well as the number of darts and hops through duration of test. Males were scored for total number of mounts and intromissions.

## Measurement of placental scars

Post-partum mothers were sacrificed 1-day post-partum. The abdominal cavity was opened and both uterine horns gently removed. Placental scars were identified as distinctive dark brown spots, counted, and logged (*Conaway and Conaway, 1955*).

## Virus preparation

The viral vector pLenZs-tetOFF-BFP-shRNAmir-HygR was redesigned based on the backbone of pGIPZ vector originally from Open Biosystems to implement the new features and better single restriction enzyme cutting sites for molecular cloning. Briefly, PCR products for tetOFF and its response elements (TetOff Gene Expression System from Clontech, Mountain View, CA), tagBFP (pTagBFP-H2B vector from Evrogen, Farmingdale, NY), and a hygromycin-resistant gene (pSilencer-hygro vector from Ambion, Grand Island, NY) were inserted in to the original pGIPZ vector to replace the unwanted components (e.g., original fluorescent protein and the puromycin resistant gene). The constructed vector map is shown in *Figure 2D*. To construct the shRNA against RFRP, a 22 nucleotide-mer oligo against RFRP gene was designed using the online program maintained by Dr Ravi Sachidanandam's Lab (http://katahdin.mssm.edu/siRNA/RNAi.cgi?type=shRNA). The oligo was inserted into the linearized pLenZs-tetOFF-BFP-shRNAmir-HygR vector using KpnI and EcoRI enzymes after adding enzyme arms on both sides of the oligo using PCR. Lentiviral particles were prepared by PEG-2000 purification of transfected Hek-293 cells and concentrated to titers of $10^9$–$10^{10}$ infectious particles per ml. The control virus was a non-silence vector commercially available from Open Biosystems (Lafayette, CO), with similar GC content and BLASTed to verify non-specificity.

  RFRP sequence: CACAGCAAAGAAGGTGACGGAA.
  Control sequence: CTCTCGCTTGGGCGAGAGTAAG.

## Stereotaxic surgery

Stereotaxic microinjections of the RFRP-shRNA and scrambled control viral particles were injected in the hypothalamus as described previously (*Goosens and Maren, 2001*). Coordinates for viral injection into the dorsal medial hypothalamus were: −3.3 mm anterior/posterior, ±0.5 mm medial/lateral relative to bregma, −8.4 mm relative to dura with skull level between bregma and lambda. Virus was infused at a rate of 0.2 µl/min for 5 min (1 µl total). At 6–8 hr after surgery, all rats received an injection of meloxicam (2 mg/kg, s.c.).

## Immunohistochemical staining for virus verification

One series of free-floating sections were rinsed in 0.1M PBS then incubated in 0.3% $H_2O_2$ in PBS for 10 min. After rinsing, tissue was blocked with 2% normal donkey serum, 0.3% Triton-X 100 in PBS, then transferred into primary antibody against GnIH (PAC123/124, Bentley) 1:5000 in PBS plus 0.3% Triton-100 [PBS-T] and sections were incubated in antibody overnight, on rotation, at 4°C. The next day, sections were rinsed in PBS and incubated in secondary for 1 hr at room temperature (Biotin donkey anti-rabbit 1:500, Jackson ImmunoResearch, West Grove, PA). Following rinsing, sections were incubated in ABC reagent (Vector) and then amplified by incubating in biotinylated tyramide for 30 min. Tertiary incubation for 1 hr at room temperature followed with streptavidin-Alexa594 (1:1000 in PBS, Jackson Immunoresearch). Following tertiary incubation, sections were incubated in an antibody against blue-fluorescent protein (anti-BFP; 1:5000, Abcam, Cambridge, MA) on a rotating stage, overnight, at 4°C. The next day, sections were rinsed in PBS then incubated in secondary antibody for 2 hr at room temperature (donkey anti-rabbit cy5, Jackson Immunoresearch). After rinsing in PBS-T, slides were coverslipped using DABCO antifading medium and stored in the dark at 4°C.

## Real-time reverse transcriptase PCR

Real-time reverse transcriptase PCR was run on TRIzol-extracted RNA further purified with DNase (DNA-free, Ambion). Rat primers were designed using NCBI Primer BLAST software, which verifies specificity. The Ct values were determined using PCR miner (*Zhao and Fernald, 2005*) and normalized to the ribosomal reference gene, ribosomal protein L16P (RPLP). There were no significant differences in RPLP values across any groups. For all studies, two-step PCR was used, following the manufacturer's instructions for iScript cDNA synthesis kit (BioRad, Hercules, CA) and then the manufacturer's instructions

for SsoAdvanced SYBR supermix (BioRad). Samples were run in a BioRad CFX96 real-time PCR system. After the PCR was complete, specificity of each primer pair was confirmed using melt curve analysis, and all samples run on a 2% ethidium bromide agarose gel with a 50-bp DNA ladder (Invitrogen, Carlsbad, CA) to verify correct product size.

Primer sequences:

| Primer | Forward | Reverse | Temp | Product Size |
|---|---|---|---|---|
| RPLP | ATCTACTCCGCCCTCATCCT | GCAGATGAGGCTTCCAATGT | 55 | 159 |
| RFRP | CCAAAGGTTTGGGAGAACAA | GGGTCATGGCATAGAGCAAT | 55 | 110 |
| GPR147 | GGTCAGAACGGGAGTGATGT | AGGAAGATGAGCACGTAGGC | 55 | 119 |
| LHβ | GCAAAAGCCAGGTCAGGGATAG | AGGCCCACACCACACTTGG | 55 | 92 |
| FSHβ | TTCAGCTTTCCCCAGGAGAGATAG | ATCTTATGGTCTCGTACACCAGCT | 55 | 305 |
| TSHβ | TCGTTCTCTTTTCCGTGCTT | CGGTATTTCCACCGTTCTGT | 55 | 245 |
| glycoprotein alpha subunit | CTATCAGTGTATGGGCTGTTG | CTTGTGGTAGTAACAAGTGC | 55 | 199 |
| KISS1 | TGGCACCTGTGGTGAACCCTG | ATCAGGCGACTGCGGGTGGCA | 61.4 | 202 |
| GnRH | GCAGATCCCTAAGAGGTGAA | CCGCTGTTGTTCTGTTGACT | 55 | 201 |

## Statistical analysis

In the chronic stress and reproductive success experiments, group differences in reproductive success, mating success, and pregnancy success were examined using G-statistics and Fisher's exact tests on raw numbers, not percentages. Litter size, placental scar, embryo survival, estradiol measurements, lordosis quotient, and intensity differences were assessed using two-way analysis of variance (ANOVA) followed by Bonferroni post-hoc tests. Differences in genes examined via RT-PCR were analyzed by a Kruskal–Wallis one-way ANOVA followed by Dunn's multiple comparison test for post-hoc analysis. Differences in corticosterone concentrations were subjected to repeated two-way ANOVAs followed by Bonferroni post-hoc test to determine statistical differences. $*p < 0.05$, $**p < 0.01$, $***p < 0.001$. Statistics were performed using R (for G-statistics and Fisher's exact test) and Prism software.

## Acknowledgements

We thank Dr I Zucker, D Covarrubias, AR Friedman, and ED Kirby for critical discussion and editorial comments, M Krolikoski and S Sanger for help in data collection, the Berkeley Molecular Imaging Center and H Aaron for imaging help. The University of Virginia Center for Research in Reproduction Ligand Assay and Analysis Core is supported by the Eunice Kennedy Shriver NICHD/NIH (SCCPIR) Grant U54-HD28934.This work was supported by NIMH BRAINS R01 MH087495 (DK) and by NIH R01 HD050470 (LJK).

## Additional information

### Funding

| Funder | Grant reference number | Author |
|---|---|---|
| National Institute of Mental Health | BRAINS R01 MH087495 | Daniela Kaufer |
| Foundation for the National Institutes of Health | R01 HD050470 | Lance J Kriegsfeld |

The funders had no role in study design, data collection and interpretation, or the decision to submit the work for publication.

### Author contributions

ACG, Conception and design, Acquisition of data, Analysis and interpretation of data, Drafting or revising the article, Contributed unpublished essential data or reagents; SEM, Conception and

design, Acquisition of data, Drafting or revising the article; SZ, Drafting or revising the article, Contributed unpublished essential data or reagents; GEB, LJK, DK, Conception and design, Analysis and interpretation of data, Drafting or revising the article, Contributed unpublished essential data or reagents

## Author ORCIDs

Sandra E Muroy, http://orcid.org/0000-0002-4014-5189

## Ethics

Animal experimentation: All animal procedures were approved by the UC Berkeley Animal Care and Use Committees (Protocol R303-0313BC) and performed in accordance with the recommendations in the Guide for the Care and Use of Laboratory Animals of the National Institutes of Health. All surgery was performed under isoflurane anesthesia with appropriate pre and post-operative analgesia, and every effort was made to minimize suffering.

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
