## [Decision Letter]

Thank you for sending your work entitled “Knockdown of a single gene prevents stress-induced infertility and embryo resorption” for consideration at *eLife*. Your article has been favorably evaluated by a Senior editor, a Reviewing editor, and 2 reviewers.

The Reviewing editor and the two reviewers discussed their comments before we reached this decision, and the Reviewing editor has assembled the following comments to help you prepare a revised submission.

This study is novel and interesting, and could potentially have significant impact. There are some concerns, however, and the questions listed below need to be addressed:

1) Knowing the levels of reproductive hormones, especially E2 and progesterone, is critical for interpreting pregnancy measures and sexual behavior, not to mention mRNA expression of multiple genes measured, and could potentially alter how the current data are interpreted. Stress can alter E2 levels and E2 can down-regulate RFRP expression in rodent hypothalamus (Molnar et al, 2011, Endocrinology; Poling et al, 2012, Endocrinology), strongly alters KISS expression (Smith et al., 2005, Endocrinology), and is essential for normal female sexual behavior and hence, successful fertility (Pfaff, 1973, Science; Rissman et al, 1997, Endocrinology). Decreases in circulating serum LH values are frequently reported to demonstrate that HPG axis is impaired by stress treatments (including in the context of RFRP; Kirby et al. 2009, PNAS). Without knowing circulating E2 or LH values, it is difficult to interpret the current experimental paradigm and outcomes. Could reproductive behavioral deficits or reversals reflective of changes in E2 levels? Is the increase in RFRP seen after stress a secondary effect due to stress's down-regulation of E2, which will result in a removal of E2 inhibition on RFRP expression? Is the neuroendocrine reproductive axis functionally active during this chronic stress paradigm (e.g. are estrous cycles normal during or after the 18 days of stress)? Are serum E2 and LH levels reversed in stressed RFRP knockdown rats? Is progesterone altered during pregnancy by the stress paradigm or prior RFRP knockdown?

2) It is not clear what value or significance the pituitary data in Figure 2 add, especially since many of the study's reproductive measures (mating, pregnancy, embryo survival, etc.) are not directly regulated by gonadotropins but only indirectly influenced due to ovarian sex steroid secretion. Additionally, most of the pituitary genes did not change (except during estrous in a few cases), making the functional significance unclear. Even in the few cases where pituitary genes changed, it is not certain these are functionally relevant changes, since serum LH, FSH, or E2 levels are not reported. LH release is itself regulated separately from transcription and does not always reflect mRNA levels (further necessitating the measure of serum LH levels).

3) The authors may need to determine if the recovery from “RFRP knockdown” to “normal elevated RFRP” itself alters reproductive success, independent of stress history. A possible interpretation of the authors' data is that RFRP add-back, via DOX suppression of the shRNA, has an unexpected positive effect on fertility, which is counteracting any negative effects of prior stress, rather than the interpretation that RFRP knockdown during stress is the important factor. This is supported by the findings that reproductive success is 64% in “unstressed, scrambled controls,” but 87% in “unstressed shRNA knockdown-then add back“ rats. This is a 23% difference, which is not unsubstantial. Likewise, pregnancy success was 15% higher in control rats with previous RFRP knockdown relative to control animals without prior RFRP knockdown.

4) The authors need to provide some data about how their stress paradigm affects pre-mating reproductive measures, as well as greater discussion as to how changes in RFRP's actions are possibly transduced to influence mating and fertility many days later. Did females cycle or ovulate normally during or after the stress period? It is inferred that cycles were normal since females were reported to be in all phases of the estrous cycle, but cyclicity data was not shown. It would also seem that ovulation was normal, assuming no differences in # of placental scars between groups (this data was not reported but should be). If cycles, ovulation, and circulating gonadotropins and E2 are normal, then females are either not stressed or estrous cycles and ovulation are resistant (or habituate) to repeated immobilization stress whereas mating and pregnancy physiology are more stress-sensitive. If so, what's the possible mechanism or model for how stress or early changes in RFRP well before mating persist to influence pregnancy later on when CORT and RFRP are back to normal levels? Particularly in regards to pregnancy and copulation success, is RFRP acting on mating neural circuits? Or pregnancy physiology? Is embryo resorption a stress-induced uterine or ovarian pregnancy hormone issue influenced somehow by brain RFRP signaling?

5) The authors conclude that the stress-induced dysfunction in reproduction is solely the effect of RFRP3, however, there is much evidence that kisspeptin (KISS) is also greatly down regulated) by stress (e.g., Kinsey-Jones et al, 2008, J Neuroendocrinology). Yet, the authors report no changes in KISS. This contradictory outcome is not properly discussed. Additionally, there is a limitation with the KISS measures in that the hypothalamic chunk contains several distinct KISS populations, which often show differential responses to treatments, and play different functional roles in fertility. Combining the separate KISS populations together in one measure diminishes interpretability.

6) Considering that there was no effect of chronic stress on lordosis quotient or lordosis intensity (4I and 4J), the dysfunction presented in 4C and 4D may be the fault of the partner breeder male, not the female. Lack of intromissions with a lordotic female would be suggestive of male, not female, sexual dysfunction.

7) In Figures 1 and 2, in each graph the data should all be plotted be relative to diestrus control, not relative to control animals within each separate cycle stage, since the latter way abolishes cycle changes in gene expression, which are themselves important to assess with and without stress.

8) There are some concerns over the use of the proper statistics/analyses in several cases.

a) A 1-way ANOVA is inappropriate for determining whether stress (vs. control) responded differently in the shRNA vs. scrambled animals. This comparison warrants use of 2-way ANOVA to determine if there is a significant interaction between treatments (shRNA vs. Scrambled) and group (control vs. Stress).

b) In Figures 1 and 2, the control animal error bars appear to be identical and plotted twice, for both stress and post-stress. Was there only one control group (if so, killed on what day?) or were there two control groups (stressed and 4d recovery) that were then combined? In either case, what is the rationale for not using separate control groups for each scenario, stressed and post-stress recovery?

c) “Unsuccessful maters” where thrown out of the study, but no information is given on this. Which group(s) were these animals from? Why were these rats not included in the lordosis quotient and lordosis intensity (with scores of 0, as would be appropriate for animals that did not show lordosis)? These animals are potentially an important aspect of the analysis.

---

## [Author Response]

1) Knowing the levels of reproductive hormones, especially E2 and progesterone, is critical for interpreting pregnancy measures and sexual behavior, not to mention mRNA expression of multiple genes measured, and could potentially alter how the current data are interpreted. Stress can alter E2 levels and E2 can down-regulate RFRP expression in rodent hypothalamus (Molnar et al, 2011, Endocrinology; Poling et al, 2012, Endocrinology), strongly alters KISS expression (Smith et al., 2005, Endocrinology), and is essential for normal female sexual behavior and hence, successful fertility (Pfaff, 1973, Science; Rissman et al, 1997, Endocrinology). Decreases in circulating serum LH values are frequently reported to demonstrate that HPG axis is impaired by stress treatments (including in the context of RFRP; Kirby et al. 2009, PNAS). Without knowing circulating E2 or LH values, it is difficult to interpret the current experimental paradigm and outcomes. Could reproductive behavioral deficits or reversals reflective of changes in E2 levels? Is the increase in RFRP seen after stress a secondary effect due to stress's down-regulation of E2, which will result in a removal of E2 inhibition on RFRP expression? Is the neuroendocrine reproductive axis functionally active during this chronic stress paradigm (e.g. are estrous cycles normal during or after the 18 days of stress)? Are serum E2 and LH levels reversed in stressed RFRP knockdown rats? Is progesterone altered during pregnancy by the stress paradigm or prior RFRP knockdown?

In response to the reviewers’ request we measured E2 and LH. In an effort to minimize additional stressors to the females in addition to the immobilization stress that occurred prior to mating and pregnancy, blood samples were only taken during the stress exposure at 9am (before the stressor began) and at 12pm (when the stressor was terminated) and not taken from controls or pregnant dams. This limits our ability to measure reproductive hormones in the present study at other time points, precluding measurements of progesterone in pregnant dams. We were able to analyze levels of LH and E2 from tail bleed samples taken throughout the stressor. The LH blood levels peak in Sprague-dawley females about 3 hours before lights out (5pm) and are low throughout the rest of the cycle. Accordingly, LH measurements in most samples fell below the detection range and did not differ among conditions.

However, we were able to measure E2 in stressed animals across all cycle stages and compare treatment with scrambled or RFRP shRNA virus, as well as compare animals that successfully mated with those that did not. These new and exciting data are now added to Figure 4. In short, we found that stressed animals that received RFRP shRNA virus had significantly higher serum estradiol levels in the morning of proestrus than stressed animals that received the control scrambled virus, implying the RFRP knockdown reverses the stress-induced blockade of the E2 rise that occurs during proestrus. Analysis of proestrus E2 levels by mating success revealed that animals that received RFRP shRNA and successfully mated had significantly higher estradiol levels than both successful and unsuccessful maters in the control scrambled groups. The data are now added to Figure 3—figure supplement 1, described in the Results section, and interpretation of the results in light of these additional data is added to the Discussion.

*Is the neuroendocrine reproductive axis functionally active during this chronic stress paradigm (e.g*. *are estrous cycles normal during or after the 18 days of stress)?*

The estrous cycle continues normally during and after the 18 days of stress. This observation has now been added to the manuscript to read: “Rats were monitored daily by vaginal smear to determine whether estrous cyclicity was affected during application of the stressor and to allow separation of animals into different cycle stages (diestrus, proestrus and estrus) at the termination of the stressor. Stress did not affect estrous cyclicity, with all animals exhibiting normal vaginal cytology throughout the stressor.”

*2) It is not clear what value or significance the pituitary data in*
Figure 2
*add, especially since many of the study's reproductive measures (mating, pregnancy, embryo survival, etc.) are not directly regulated by gonadotropins but only indirectly influenced due to ovarian sex steroid secretion. Additionally, most of the pituitary genes did not change (except during estrous in a few cases), making the functional significance unclear. Even in the few cases where pituitary genes changed, it is not certain these are functionally relevant changes, since serum LH, FSH, or E2 levels are not reported. LH release is itself regulated separately from transcription and does not always reflect mRNA levels (further necessitating the measure of serum LH levels)*.

We have removed all the pituitary data.

*3) The authors may need to determine if the recovery from “RFRP knockdown” to “normal elevated RFRP” itself alters reproductive success, independent of stress history. A possible interpretation of the authors' data is that RFRP add-back, via DOX suppression of the shRNA, has an unexpected positive effect on fertility, which is counteracting any negative effects of prior stress, rather than the interpretation that RFRP knockdown during stress is the important factor. This is supported by the findings that reproductive success is 64% in “unstressed, scrambled controls,” but 87% in “unstressed shRNA knockdown-then add back” rats. This is a 23% difference, which is not unsubstantial. Likewise, pregnancy success was 15% higher in control rats with previous RFRP knockdown relative to control animals without prior RFRP knockdown*.

The difference (e.g. 64% vs. 87%) was non-significant though we understand the reviewer’s concern. We suspected that this difference may be due to the smaller number of animals in the control group (that resulted from a cage removed from the study because of fighting wounds) and not reflecting a biological difference. To test this possibility, we ran an additional experiment: control animals received either RFRP shRNA or control scrambled virus via stereotaxic injections in the hypothalamus (n=12 total). Copulation, pregnancy success and embryo survival were measured in this cohort. The new data were added to the dataset in Figure 3, and they demonstrate that reproductive success is not different in the control group treated with RFRP shRNA compared to control scrambled. (Reproductive success: 76 vs. 80%; pregnancy success: 87 vs. 89%; Copulation success: 88 vs. 90%; Embryo survival: 98 vs. 98%).

4) The authors need to provide some data about how their stress paradigm affects pre-mating reproductive measures, as well as greater discussion as to how changes in RFRP's actions are possibly transduced to influence mating and fertility many days later. Did females cycle or ovulate normally during or after the stress period? It is inferred that cycles were normal since females were reported to be in all phases of the estrous cycle, but cyclicity data was not shown. It would also seem that ovulation was normal, assuming no differences in # of placental scars between groups (this data was not reported but should be). If cycles, ovulation, and circulating gonadotropins and E2 are normal, then females are either not stressed or estrous cycles and ovulation are resistant (or habituate) to repeated immobilization stress whereas mating and pregnancy physiology are more stress-sensitive. If so, what's the possible mechanism or model for how stress or early changes in RFRP well before mating persist to influence pregnancy later on when CORT and RFRP are back to normal levels? Particularly in regards to pregnancy and copulation success, is RFRP acting on mating neural circuits? Or pregnancy physiology? Is embryo resorption a stress-induced uterine or ovarian pregnancy hormone issue influenced somehow by brain RFRP signaling?

According to our cytology measurements (explained above), females continue to cycle normally throughout the stress procedure, although RFRP levels are elevated throughout and E2 levels are low. To be included in the study, animals had to exhibit normal vaginal cytology during the 4-5 day estrous cycle after the stress ended.

Additionally, we have now added the data reporting no difference in the number of placental scars (Figure 2).

The following section has been added to our discussion to consider a possible mechanism: “The stress-induced rise in RFRP may be acting on neural circuits influencing mating and pregnancy, potentially independently of sex steroids. RFRP projects to multiple brain regions responsible for successful reproduction and mating behavior, including the medial preoptic area (mPOA) (where it is known to affect GnRH release) as well as the BNST, medial amygdala, anterior hypothalamus and arcuate nucleus (12). Piekarski et. al. found that administering RFRP3 to hamsters reduced sexual motivation (as measured by percent of time spent with castrated vs. intact males) and vaginal scent marking without effect on lordosis behavior, similar to our present findings. Additionally, RFRP3 administration altered cellular activation in regions of the brain implicated in female sexual motivation, including the mPOA, medial amygdala, and BNST - all regions that receive RFRP projections. These effects were independent of gonadal steroids and kisspeptin cellular activation (4). While we were unable to measure progesterone or prolactin in this study, it is possible that RFRP projections to the arcuate nucleus affect dopaminergic signaling required for prolactin release and maintenance of progesterone levels and pregnancy success. Future studies aimed at systematically examining each step in these processes is required to gain a full understanding of the neural circuits underlying the deleterious effects of stress on reproduction.”

*5) The authors conclude that the stress-induced dysfunction in reproduction is solely the effect of RFRP3, however, there is much evidence that kisspeptin (KISS) is also greatly down regulated) by stress (e.g., Kinsey-Jones et al, 2008, J Neuroendocrinology). Yet, the authors report no changes in KISS. This contradictory outcome is not properly discussed. Additionally, there is a limitation with the KISS measures in that the hypothalamic chunk contains several distinct KISS populations, which often show differential responses to treatments, and play different functional roles in fertility. Combining the separate KISS populations together in one measure diminishes interpretability*.

As this reviewers indicate, the KISS measurements are difficult to interpret as anterior and posterior cell populations cannot be disambiguated. It is possible that a change in one population, without an accompanying change in the other, masks a significant effect. As a result, we cannot rule out a role for kisspeptin downstream of RFRP. This consideration is now discussed in the manuscript text. However, suppression of RFRP during stress was sufficient to rescue the impact of stress on reproductive outcomes, underscoring the role of RFRP (vs. kisspeptin involvement). These points are now clarified in the Discussion of the revised manuscript.

*6) Considering that there was no effect of chronic stress on lordosis quotient or lordosis intensity (4I and 4J), the dysfunction presented in 4C and 4D may be the fault of the partner breeder male, not the female. Lack of intromissions with a lordotic female would be suggestive of male*, *not female, sexual dysfunction.*

In the original figure we presented the behavioral data (lordosis quotient or lordosis intensity) from the successful maters only, showing no differences between groups. Based on the reviewer suggestion to score non-maters as 0, we have now changed the figure to include unsuccessful maters with the successful maters for both lordosis quotient and intensity (new Figure 3—figure supplement 1), and results described in the text. We found a significant main effect of stress on lordosis intensity, though post hoc tests showed no significant differences within groups. Lordosis quotient measures revealed significantly lower ratio in the scrambled stress group compared to the nonstressed groups that received scrambled or RFRP-shRNA (0.30±0.10 vs. 0.73±0.07 and 0.68±0.07), indicating that stress exposure decreased the relative sexual receptivity of the females (Figure 3—figure supplement 1), congruent with the stress-induced drop in mating success. Interestingly, LQ ratios in stressed females that received RFRP-shRNA were not significantly different from controls ratios (0.53±0.10, Figure 3—figure supplement 1), demonstrating that knockdown of RFRP reversed the stress-induced decrease in sexual receptivity, and congruent with the reversal of mating success found in this group.

*7) In*
Figures 1 and 2*, in each graph the data should all be plotted be relative to diestrus control, not relative to control animals within each separate cycle stage, since the latter way abolishes cycle changes in gene expression, which are themselves important to assess with and without stress*.

Figure 1 data has all been reevaluated so that each group (18D and 18D+4) is analyzed relative to diestrus control. The only case is which this markedly changed the original data is found in Figure 1, where we no longer see an increase in the RFRP receptor during proestrus. GnRH appears lower in proestrus animals after stress, however this difference is not significant.

*8) There are some concerns over the use of the proper statistics/analyses in several cases*.

*a) A 1-way ANOVA is inappropriate for determining whether stress (vs. control) responded differently in the shRNA vs. scrambled animals. This comparison warrants use of 2-way ANOVA to determine if there is a significant interaction between treatments (shRNA vs. Scrambled) and group (control vs. Stress)*.

We thank the reviewers for noting this oversight that has now been remedied. Data is now analyzed using a 2-way ANOVA, and the Methods, Results and figure legend has been edited to reflect this change.

*b) In*
Figures 1 and 2*, the control animal error bars appear to be identical and plotted twice, for both stress and post-stress. Was there only one control group (if so, killed on what day?) or were there two control groups (stressed and 4d recovery) that were then combined? In either case, what is the rationale for not using separate control groups for each scenario, stressed and post-stress recovery?*

Originally, all groups (18D and 18D+4) were shown on the same graph and controls were collapsed as they were not statistically different. When they were separated into different graphs, the control groups were not appropriately separated. We thank the reviewer for identifying this error. We re-plotted these graphs each with their own controls.

*c) “Unsuccessful maters” where thrown out of the study, but no information is given on this. Which group(s) were these animals from? Why were these rats not included in the lordosis quotient and lordosis intensity (with scores of 0, as would be appropriate for animals that did not show lordosis)? These animals are potentially an important aspect of the analysis*.

Unsuccessful maters were found in all groups: sample sizes have been incorporated into the figure legend to indicate where animals were removed from each step. Lordosis quotient and intensity has been moved to Figure 3—figure supplement 1, and now includes totals of all animals, rather than separated by mating. (See answer to comment 6.)